# Integration of Ultra- and Nanofiltration for Potato Processing Water (PPW) Treatment in a Circular Water Recovery System

**DOI:** 10.3390/membranes13010059

**Published:** 2023-01-03

**Authors:** Paulina Rajewska, Jolanta Janiszewska, Jakub Rajewski

**Affiliations:** 1Łukasiewicz Research Network—Institute for Sustainable Technologies, ul. K. Pułaskiego 6/10, 26-600 Radom, Poland; 2Faculty of Chemistry, Warsaw University of Technology, Noakowskiego 3, 00-664 Warszawa, Poland; 3Łukasiewicz Research Network—New Chemical Syntheses Institute, Al. Tysiąclecia Państwa Polskiego 13a, 24-110 Puławy, Poland

**Keywords:** ultrafiltration, nanofiltration, potato processing water

## Abstract

The article analyzes integrated ultrafiltration (UF) and nanofiltration (NF) processes for potato processing wastewater treatment for the purpose of which a laboratory filtration system for flat sheet membranes with the effective surface area of 1.4 × 10^−2^ m^2^ (UF: polysulfone, cut-off: 10,000 Da; NF: polypiperazine amide, cut-off: 150–300 Da) was used. As part of the study, the effect of the transmembrane pressure of UF (0.2 MPa and 0.4 MPa) and NF (1.0 MPa and 1.8 MPa) on the permeate flux and rejection coefficient was investigated and the impact of sewage preparation methods on the degree of pollution reduction was determined. Moreover, a method for a fouling layer removal from the UF membranes is also proposed. The results of the analyses conducted by the authors show that the pretreatment stage offers additional advantages to TSS and turbidity removal. In both cases (0.2 and 0.4 MPa), UF used after the pretreatment process resulted in a 97–99% reduction in these impurities. The analysis of the determined rejection coefficients shows that the use of NaOH and H_2_O_2_ for the regeneration of the UF membrane has a positive effect on filtration efficiency. Regarding NF, the rejection coefficients for most tested parameters were higher for the 1.8 MPa process compared to 1.0 MPa, and approximately 80% of water was recovered.

## 1. Introduction

Potatoes are the leading non-cereal crop in the world and the fourth most important crop after rice, wheat, and maize [1]. Potatoes were cultivated on 1.7 million hectares (ha) in the EU-27 in 2020. This corresponded to an estimated 1.7% of all arable land in the EU. The harvested production of potatoes in the EU was 55.3 million tons in 2020. Germany was the largest producer of potatoes in the EU in 2020 (at 11.7 million tons, 21.2% of the EU total), ahead of Poland (a provisional 16.4%), France (15.7%) and the Netherlands (12.7%) (Figure 1). A typical potato food industry generates between 900 and 1700 m^3^ of wastewater per day [2].

Potatoes contain approximately 77.2% water, 2.7% protein, 0.3% fat, 1% ash, 18.8% carbohydrates, and 1.0% reducing sugars [3]. Therefore, as many authors note, the post-process water generated in the processes of potato cutting and washing (Figure 2 [1]) contains significant amounts of suspension (mainly starch), proteins, and other impurities.

Companies active in the potato sector process ca. 13,600 tons of raw potatoes to produce chips and dried potatoes and for that purpose, use approximately 230 million liters of water. The amount of waste generated during potato processing, i.e., washing, cutting, or hydrotransport, can be compared to the amount of waste generated by a city with 200,000 inhabitants [4].

Due to the above-mentioned factors, the PPW cannot be discharged directly into the sewage system or the environment. It is also significantly challenging to reuse the PPW in production (mainly due to the high chemical and biological demand for oxygen and suspended solids). PPW treatment and reuse are often discussed in the literature and involve the implementation of complex, multi-stage treatment processes [5].

Multi-stage wastewater treatment usually consists of three phases: primary (pre-treatment), secondary, and tertiary. The use of flotation, sedimentation, or sieving as a pre-treatment method enables the removal of suspended solids and deposited solids. Remains of the dissolved organic particles are then removed in step 2 (secondary treatment) in biological processes, which can be natural or mechanically assisted. The main disadvantage of the applied methods is the long process time. As a result, large-volume bioreactors allowing industrial effluent treatment are needed. Additionally, microorganisms are very sensitive to such factors as temperature and pH. Therefore, tertiary treatment is often used, which may include the following operations: microstraining, granular media filtration, chemical coagulation, nitrification-denitrification, air stripping, and ion exchanging, which usually employ membrane technologies (reverse osmosis and ultrafiltration) [6,7,8].

Generally, pressurized membrane techniques are successfully used to treat industrial wastewater. Membrane filtration enables the recovery and recirculation of raw materials and water in the production process. In addition, membrane techniques offer the advantages of simple handling and lower operating costs compared to conventional technologies. These features make membrane filtration widely used in the food industry [9,10]. Membrane filtration is classified in BAT (Best Available Technology) as a waste-free technology. However, due to technical problems related to fouling, membrane filtration is not commonly used for the purification of PPW.

The efficiency of membrane processes is defined as the number of substances retained by the membrane (retention coefficient) or the permeation of solutes through the membrane (permeability). The work [11] presents the process parameters and the construction of the reverse osmosis installation. Wastewater from potato processing contains large amounts of starch. The ultrafiltration-reverse osmosis system can be successfully used to recover potato starch from wastewater [4].

Regardless of the wastewater type, membrane fouling is caused by the accumulation of raw sewage components on the membrane surface and in its pores. In addition, concentration polarization and gel formation—chemical interaction between the solutes and the membrane material—co-occur and microorganisms multiply [12,13]. Membrane fouling is always a problem in the practical application of pressurized membrane techniques, as it results in a decrease in the permeate flux over time. To avoid these problems, pre-treatment of wastewater (such as centrifugation, sedimentation, pre-filtration, and microfiltration) and periodic chemical washing, the so-called clean-in-place (CIP) using acids, bases, surfactants, and enzymes [14,15] are used. As reported by the authors [12,14], membrane fouling during PPW filtration is primarily from the contamination of the membrane surface with starch and proteins. Dabestani et al. [1] show that the use of an alkaline cleaning agent (e.g., NaOH) is more effective when it comes to fouling removal (97%) compared to surfactants (e.g., SDS) (83%) and acidic agents (e.g., HCl) (75%).

Given the European Commission’s adoption of the Circular Economy Package [16], which aims at the transition of EU countries to a circular economy, standard PPW treatment practices have become insufficient [17,18,19]. The adoption of restrictive environmental regulations, and thus the increase in the costs of wastewater treatment and waste disposal, results in the need to recover water and by-products (mainly starch and proteins) from PPW [1,17].

The PPW must meet the criteria laid down in relevant applicable laws. In Poland, they include the act on the conditions for the wastewater introduction into sewerage equipment; the act on the wastewater discharged into water or soil and the act on the quality of water intended for human consumption.

From such a perspective, this study intends to assess the feasibility of using a combined ultrafiltration and nanofiltration (UF/NF) membrane system for potato processing wastewater treatment. Only a few studies on the recovery of potato proteins by ultrafiltration and reverse osmosis (RO) have been reported [20,21,22,23]. Compared with UF and RO membranes, nanofiltration membranes can selectively separate monovalent salt and polyvalent salt, and effectively reject organic solutes with molecular weights above 200 Da. Ultrafiltration of water used for processing potatoes enables the reduction of its most critical physicochemical parameters. However, the reclaimed water cannot be reused in the production process as it does not meet the requirements for water intended for human consumption, nor can it be directly discharged into the environment [24].

Therefore, the combination of UF and NF can provide effective regeneration of the PPW. The aim of the work is to examine the possibility of using the UF/NF system to treat potato processing wastewater and to assess the potential of discharging the treated wastewater into the environment or reusing the recovered water in the production process. The influence of operating conditions on the efficiency of ultra- and nanofiltration is examined regarding the efficiency of removing organic compounds. In addition, a method for removing the fouling layer from UF membranes is proposed.

## 2. Materials and Methods

### 2.1. Analytical Methods

The effectiveness of water recovery after pretreatment, ultrafiltration, and nanofiltration was determined through the analysis of the following key parameters: chemical oxygen demand (COD), total organic carbon (TOC), turbidity, dry residue, total suspended soil (TSS), sulfate (VI), as well as phosphorus and nitrogen contents. All those parameters were characterized by means of spectrophotometry (DR 6000 Hach Lange, Berlin, Germany). Chemical analyses of raw wastewater, feed, retentate, and permeate were carried out using cuvette tests (Hach Lange, Germany). Characteristics of the methods used for the determination of chemical oxygen demand, total organic carbon, total phosphorous, and total nitrogen are presented in [24]. The TSS was determined using a 25 mL cuvette and measured at the 810 nm wavelength. The dry residue was determined at 105 °C using the laboratory moisture analyzer (MAC 50/1, Radwag, Radom, Poland), and turbidity was measured with a 2100Q IS Portable Turbiditymeter (Hach Lange, Germany). A Mettler Toledo (Columbus, OH, USA) SevenMulti conductometer and InLab 731 conductivity probe were used to examine changes in the conductivity of the samples collected after the filtration process.

The treatment efficiency of the investigated potato processing wastewater was determined for all tested physicochemical parameters using the retention coefficient (*R*):$$R=\left(1-\frac{x_{p}}{x_{n}}\right)\times 100\%$$
where *R*—is the retention coefficient, (%); *x_p_* and *x_n_*—are the value of the tested parameter in the permeate and the feed, respectively.

### 2.2. Characterization of the Potato Processing Wastewater (PPW)

Potato processing wastewater was obtained from a chips factory (Radom, Poland), directly from the production line after the washing and cutting process. Table 1 shows the characteristic of the raw wastewater sampled from the drain pipe located in the potato chips factory.

### 2.3. Pretreatment

Before the potato processing wastewater was subject to membrane filtration, the influence of the sewage preparation methods on the degree of pollution reduction, and thus the reduction in the membrane blocking intensity, was determined. The examined PPW was sedimented, filtered through a bag filter, or centrifuged (5–30 min) (Table 2).

### 2.4. Membrane Filtration

Membrane filtration efficiency was tested on a laboratory scale. A Sterlitech system equipped with a membrane module placed in a hydraulic press, a conical supply tank with a capacity of 19 L, a thermostatic system maintaining the constant feed temperature of 25 ± 1 °C, and a high-pressure pump (Figure 3) was used. The crossflow filtration was carried out in all membrane processes.

Flat membranes (ultrafiltration and nanofiltration) with an area of 1.4 × 10^−2^ m^2^ were used. Ultrafiltration was performed at the transmembrane pressure of 0.2 and 0.4 MPa (Table 3) [24]. The collected permeate was further purified on a nanofiltration membrane under the pressure of 1.0 and 1.8 MPa. The process was carried out in each case until the feed was exhausted.

The permeate flux (at constant temperature and pressure) was calculated with the following formula:(1)$$J_{A}=\;\frac{V}{A\cdot t}$$
where: *J_A_*—the permeate flux (mL (min cm^2^ bar)^−1^); *V*—the volume of filtrate (mL); *A*—the effective area of the flat sheet membrane (cm^2^) and *t*—the sampling time (min).

### 2.5. UF Membrane Cleaning

The possibility to regenerate UF membranes used to filtrate potato processing wastewater was investigated. The membranes used in the processes carried out at the transmembrane pressure of 0.4 MPa were regenerated. The impurities deposited on the surface and in the membrane’s pores were removed using hydrogen peroxide and sodium hydroxide solutions. The sequential procedure of soaking the dirty membrane in successive solutions was used:Deionized water (24 h),0.5% sodium hydroxide solution (6.5 h),Deionized water (16 h),0.03% hydrogen peroxide solution (6.5 h), andDeionized water (24 h).

The tested membrane regeneration procedure was compared with the effectiveness of its surface cleaning with a stream of deionized water. The cleaning evaluation efficiency procedure was carried out after three membrane filtration processes [24].

## 3. Results and Discussion

### 3.1. Pretreatment

Potato processing wastewater contains compounds of different molecular weights. Not all macromolecules sediment in a solution easily and quickly [25]. The raw industrial sewage collected from the potato-cutting process is characterized by high values of COD, suspended solids, and turbidity. It was noticed that after sedimentation and filter bag filtration, the values of the determined parameters decreased only slightly compared to centrifugation (Figure 4).

Two-hour sedimentation and filter bag filtration reduced the TSS and turbidity to a similar degree, i.e., by 23.1–25.9% and 11.5–14.1%, respectively. In the case of centrifugation, the reduction in the content of impurities in the supernatant is already significant. The use of five-minute centrifugation reduced the turbidity by 75.3% and the suspension by 28.6%. Extending the centrifugation time to 30 min allowed for a further reduction in these parameters to 92% and 61%, respectively (Figure 4). The dry residue, the total suspended and dissolved solids in all the investigated pretreatment methods was reduced from 1.1% (for the raw sewage) to approx. 0.6–0.8% (for all treated samples). In a further part of the work, five-minute centrifugation was used as a pretreatment of the investigated PPW before membrane filtration, which was a compromise between the number of pollutants removed and economic considerations.

### 3.2. Ultrafiltration

Figure 5 shows the variations of permeation water flux with operation time under 0.2 and 0.4 MPa of transmembrane pressure (TMP). Permeate flux in the ultrafiltration of PPW was (for 0.4 MPa operating pressure) higher, approximately 1.7 times, compared to 0.2 MPa.

In the filtration process carried out at the transmembrane pressure of 0.2 MPa, significantly larger amounts of only nitrogen compounds were removed and slightly larger amounts of ionic components and phosphorus compounds. At the same time, conducting the process under 0.4 MPa did not cause any significant changes in the removal of other impurities (Figure 6). The authors also found that the pretreatment stage offered additional advantages of TSS and turbidity removal. In both cases (0.2 and 0.4 MPa), UF following the pretreatment process resulted in a 97–99% reduction in these impurities. According to the data in Figure 6, UF membranes also show a high rejection of total contents of organic carbon and nitrogen (up to 80%).

Ultrafiltration of PPW carried out on membranes without periodic treatment enabled the recovery of 36–38% (*v*/*v*) of water. An increase in the flow resistance through the membrane over time and a decrease in the filtration rate due to the continuous clogging of the membrane pores and fouling were observed. This indicates that membrane fouling is unavoidable in the case of UF membranes. There are lots of potato proteins in potato processing wastewater that can easily and quickly adsorb and adhere to the porous surface of the separation membrane and as a result, block the membrane pores and cause a rapid decrease in water flux [26], as shown in Figure 5. A reduction in separation efficiency and consecutive replacement of the fouled membrane can be minimized by regular cleaning [27]. Restoring the initial filtration efficiency is impossible by simply cleaning the membrane surface with a stream of deionized water.

Fresh deionized water was not very effective for cleaning a PES membrane fouled with potato protein. Pure water removes only the proteins that adsorb on the surface of the UF membrane (i.e., reversible fouling). The proteins that block membrane micropores are difficult to remove. Since potato proteins, which are the main PPW foulants, are alkaline soluble proteins, washing with an alkaline solution was one of the steps in the sequential regeneration of the PES membrane.

The values of physicochemical parameters, i.e., dry residue, phosphorus, and total nitrogen or COD determining the purity of the collected permeate after initial processes, were significantly reduced. The use of backwashing to remove the layer of impurities on the membrane surface made of polymeric materials may cause mechanical damage to the membrane.

Polymer membranes have limited mechanical strength. Thus, it is not possible to remove the contaminant layer by reversing the water flow. Therefore, the possibility of eliminating the contamination layer from the PES membrane with sodium hydroxide and hydrogen peroxide solutions was analyzed.

The use of an appropriate configuration for removing the contamination layer of the PES membrane allowed for its effective regeneration. The efficiency of the PPW filtration process on the regenerated membrane was comparable to that of the fresh membrane (Figure 7).

After process 1, the surface of the membrane was washed with a stream of fresh deionized water. Then, the membrane was used in process 2. Comparing the permeate flux for the I and II process (Figure 7), it was noticed that removing impurities from the membrane with deionized water is difficult. A stream of fresh deionized water can only remove contaminants that have accumulated on the surface of the membrane [1]. Chemical washing can be used to remove impurities that block the pores inside. It was found that using NaOH and H_2_O_2_ to regenerate the UF membrane significantly improves the filtration efficiency of the fresh portion of PPW. The procedure of chemical cleaning was described in issue 2.5. Figure 8 presents the dependence of the retention coefficients for the physicochemical parameters of the PPW treated with UF using a new (process 1) or regenerated (2 and 3 processes) membrane. In the filtration process carried out on the regenerated membrane (III process), 159% more phosphorus compounds and 57% more nitrogen compounds were removed. Increased dry residue and COD retention rates of 72% and 51%, respectively, were also obtained. Comparable results for the physicochemical parameters were obtained for filtration on a membrane cleaned only with deionized water. In this case, however, a reduction in permeate flux was also observed. The impact of the membrane regeneration method proposed in the paper on the remaining parameters (i.e., conductivity, turbidity, TSS and TOC) was negligible.

Table 4 summarizes the physicochemical parameters and concentrations of analytes determined in the obtained filtrates (after UF and NF of PPW) with the permissible values of these parameters for water intended for human consumption, wastewater discharged into waters or the ground, and wastewater discharged into sewage systems in Poland. According to other authors [20,21], the use of the UF process alone does not allow either the reuse of the recovered water in the production process (e.g., for cleaning potatoes) or its direct discharge into the environment (water or soil). The main reason is that the turbidity of the filtrate is greater than the acceptable value. In addition, it is necessary to reduce the concentration of compounds that can be oxidized by dichromate ions (COD_Cr_), as well as the total organic carbon, phosphorus, and nitrogen content. The desired values of these parameters can be achieved by using the nanofiltration process.

### 3.3. Nanofiltration

The permeate obtained during ultrafiltration was further purified in the nanofiltration process. Figure 9 shows the changes in the permeate flux over time for nanofiltration carried out at the pressure of 1.0 and 1.8 MPa. Conducting the filtration process at higher pressure (1.8 MPa) increased the permeate flux. Moreover, due to the increased fouling effect, they resulted in an increase in filtrate flux over time. The filter cake deposited on the membrane surface and inside pores due to fouling constituted an additional permeate barrier layer. Thus, the filtrate obtained at the pressure of 1.8 MPa was characterized by a greater degree of purification (Figure 10). The rejection coefficients for most parameters tested (turbidity, TSS, COD, solid residue, sulfates (VI), total nitrogen and phosphorus) were higher for the 1.8 MPa process compared to 1.0 MPa. Water recovery in the case of NF is about 80%.

## 4. Conclusions

Ultrafiltration supported by pretreatment, as a single stage of PPW purification, does not reduce the concentration of pollutants to a level that will enable their return to the production process and reuse. The UF process can only partly remove the organic matter. Therefore, it is required to treat the UF permeation solution further. In the nanofiltration process, a significant part of ionic and organic components (high retention factors for COD and TOC), as well as phosphorus, nitrogen, and sulfate compounds, were removed in relation to ultrafiltration. The use of a coupled UF-NF system, preceded by 5-min centrifugation, made it possible to completely remove turbidity and total suspended solids, and to reduce other tested physicochemical parameters in the range of 70–99%. Unfortunately, even the use of the integrated UF-NF system did not remove a sufficient amount of organic matter to meet the necessary criteria enabling the tested wastewater to be discharged into the environment (Table 4). However, the PPW filtration system proposed in the paper can be used to close water circuits in the potato industry. The obtained filtrates meet the drinking water quality criteria applicable in Poland.

## Figures and Tables

**Figure 1 membranes-13-00059-f001:**
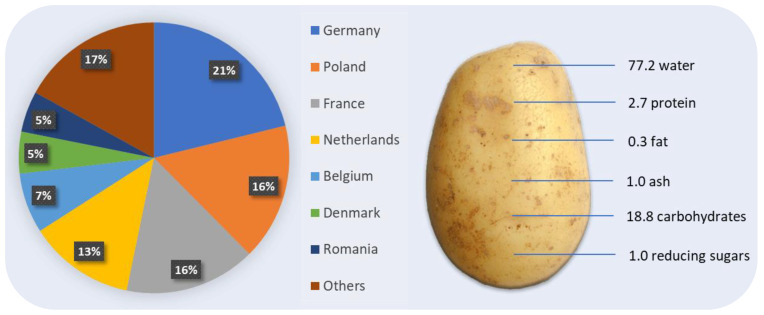
Production share of potatoes in the European Union in 2020 by a leading producer and chemical composition of a potato.

**Figure 2 membranes-13-00059-f002:**
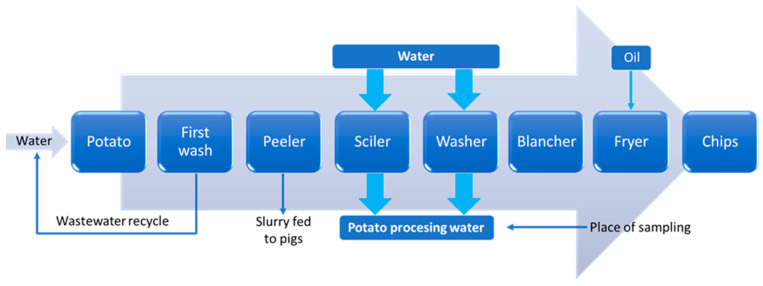
Block diagram of potato chip production and processing water.

**Figure 3 membranes-13-00059-f003:**
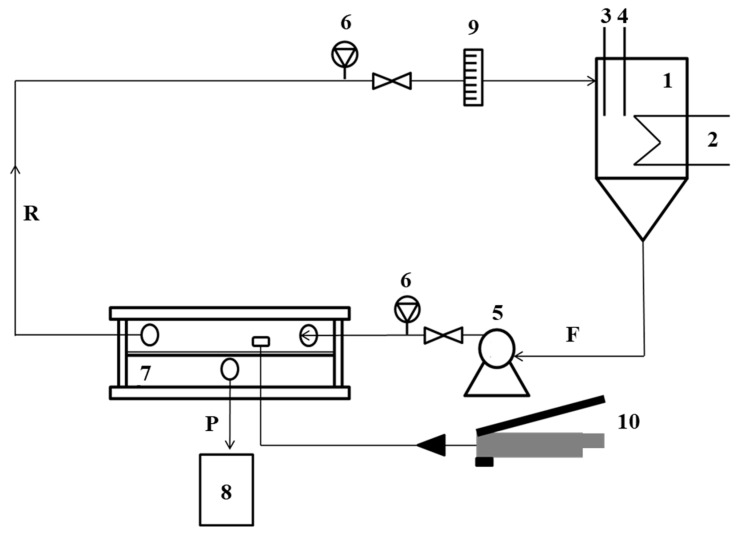
Diagram of a laboratory membrane filtration system: 1—feed tank; 2—thermostat; 3—thermometer; 4—pH-meter; 5—pump; 6—manometer; 7—filtration module; 8—permeate tank; 9—rotameter; 10—hydraulic hand pump; P—permeate; R—retentate; F—feed.

**Figure 4 membranes-13-00059-f004:**
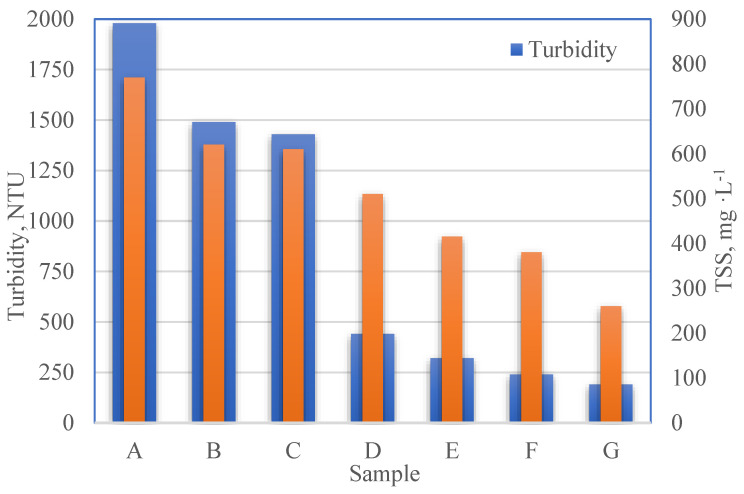
Comparison of parameter values of suspension and turbidity in the raw sewage and after the application of preliminary treatment (sedimentation, bag filter, centrifugation 5–30 min).

**Figure 5 membranes-13-00059-f005:**
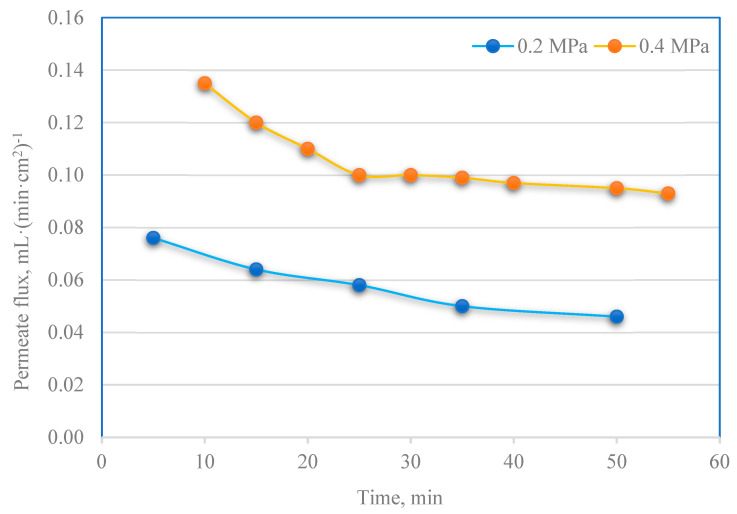
PPW permeate flux vs. time for processes conducted under 0.2 and 0.4 MPa of transmembrane pressure.

**Figure 6 membranes-13-00059-f006:**
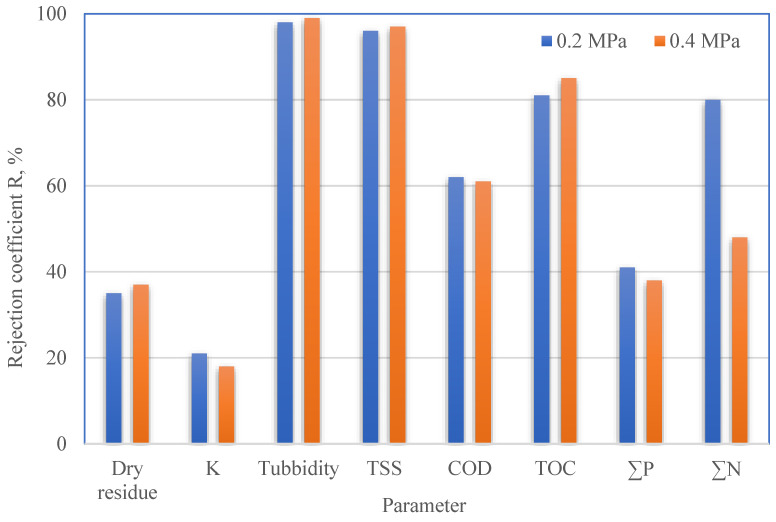
Rejection coefficients vs. physicochemical parameters of PPW purified under operating pressure of 0.2 and 0.4 MPa.

**Figure 7 membranes-13-00059-f007:**
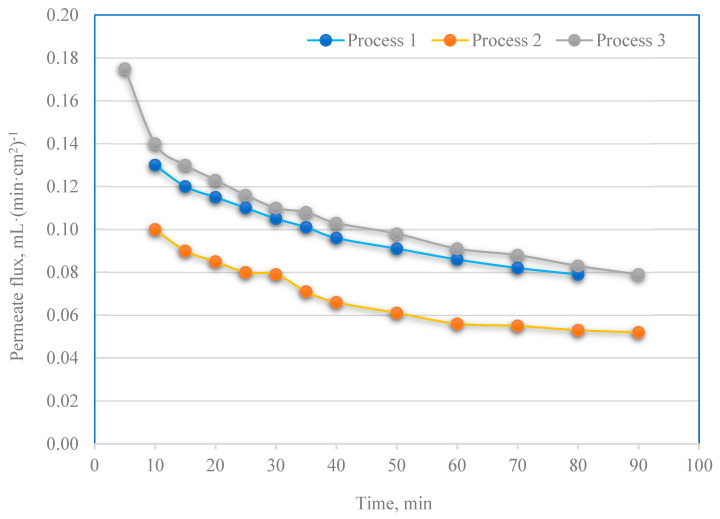
PPW permeate flux vs. time obtained on the new (process 1) and regenerated (process 2 and 3) membrane (0.4 MPa).

**Figure 8 membranes-13-00059-f008:**
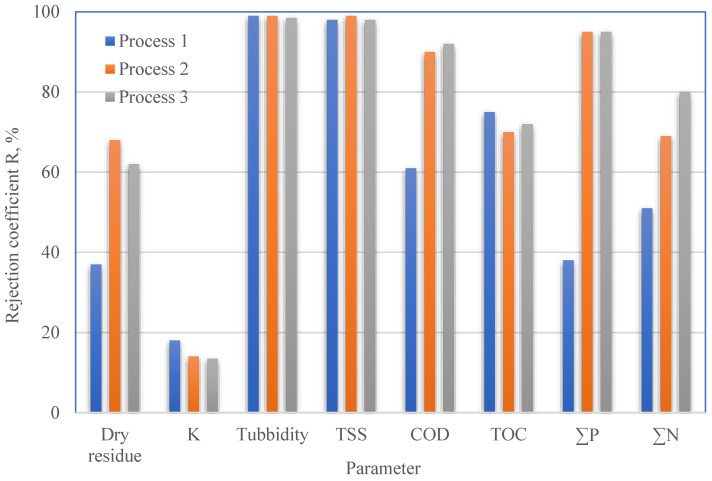
Rejection coefficients vs. physicochemical parameters of PPW treated by UF using a new (process 1) or regenerated (processes 2 and 3) membrane.

**Figure 9 membranes-13-00059-f009:**
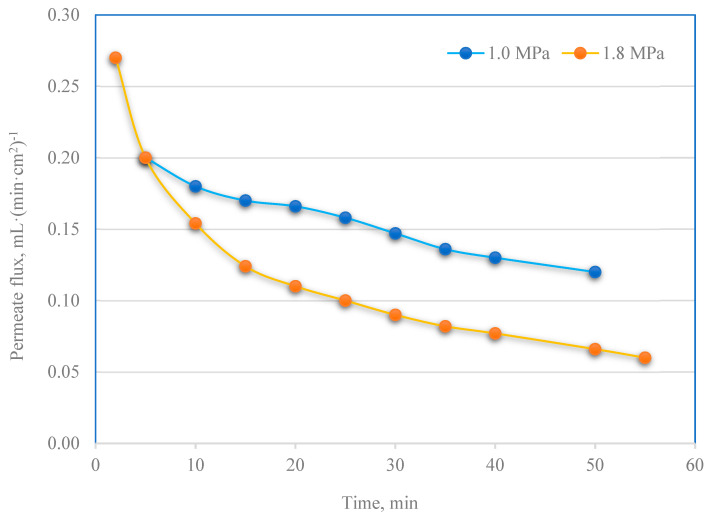
PPW permeate flux for NF processes vs. time (at 1.0 and 1.8 MPa of transmembrane pressure).

**Figure 10 membranes-13-00059-f010:**
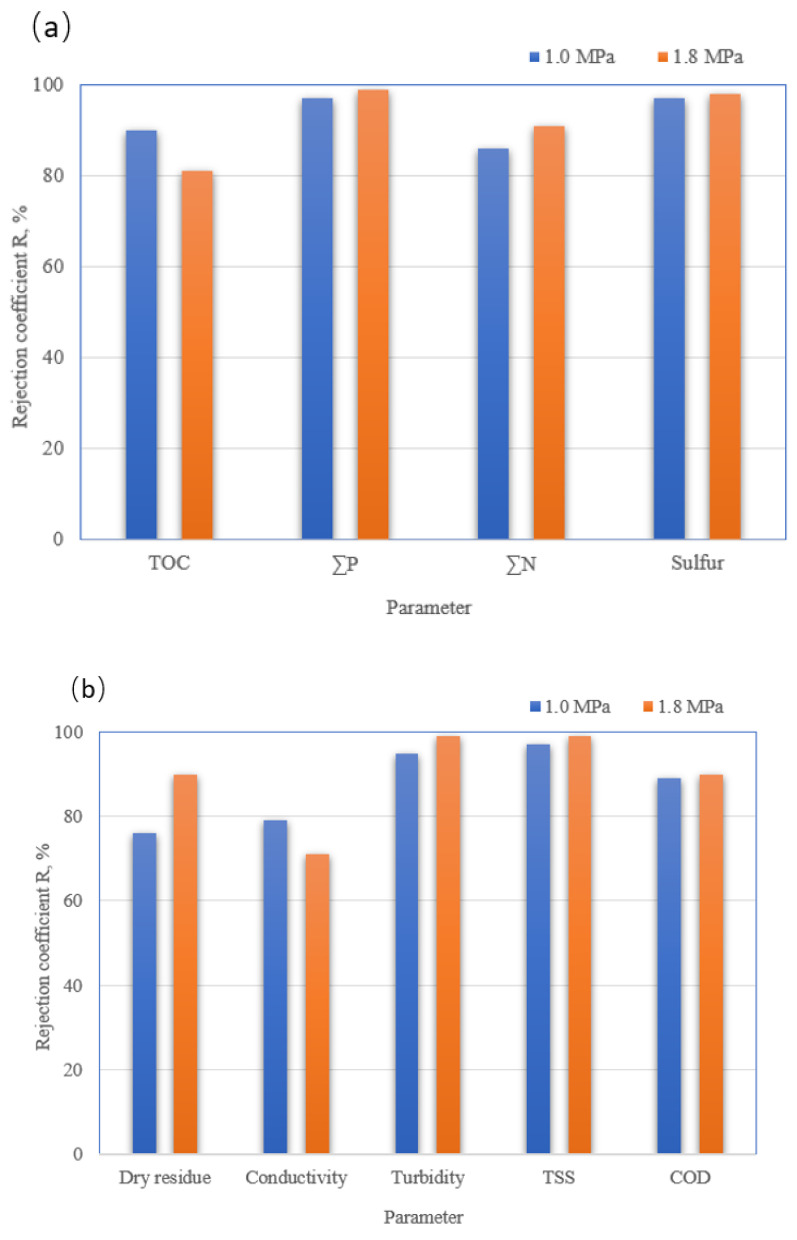
A comparison of rejection coefficients for determined physicochemical parameters: (**a**) TOC, ∑P, ∑N, and sulfur, (**b**) dry residue, conductivity, turbidity, TSS, and COD of PPW purified under operating pressure of 1.0 and 1.8 MPa.

**Table 1 membranes-13-00059-t001:** Physicochemical parameters of the examined wastewater collected directly from the production line.

Parameter/Analyte		Unit	Range of Determining Value
Dry residue		%	0.8–1.5
Conductivity		μS·cm^−1^	1187–1451
Turbidity		NTU	496–2599
Total suspended solids		mg·L^−1^	1027–1584.7
COD		mg·L^−1^	3864–9275
TOC		mg·L^−1^	846–2243
∑P (as PO_4_^3−^)		mg·L^−1^	4.0–9.17
∑N		mg·L^−1^	160.7–263.2
Microbiology	Total	pcs. mL^−1^	0.54–1.25 × 10^6^
	including G−		0.08–0.5 × 10^6^

**Table 2 membranes-13-00059-t002:** Designation of PPW samples vs. sewage treatment methods used.

Designation of the Sample	Preparation Method	Conditions
A		Raw wastewater
B	Sedimentation	2 h
C	Filtration through the bag filter	Pore diameter 5 µm
D	Centrifugation	5 min
E		10 min
F		20 min
G		30 min

**Table 3 membranes-13-00059-t003:** Characteristic of ultra- and nanofiltration processes.

	UF	NF
Membrane type	“Thin film”	“Thin film”
Polymer	polysulfone	polypiperazine amide
Cut-off [Da]	10,000	150–300
Pressure	0.2; 0.4 MPa	1.0; 1.8 MPa

**Table 4 membranes-13-00059-t004:** Comparison of the physicochemical parameters of the obtained filtrates (after UF and NF of PPW) with the allowed values of these parameters for water intended for human consumption, wastewater discharged into water or soil, and wastewater introduced into sewage equipment in Poland.

Parameter/Analyte	UF Permeate	NF Permeate	Legal Standards		
			Water Intended for Human Consumption	Wastewater Discharged into Water or Soil	Wastewater Introduced into Sewage Equipment
Dry residue/%	0.11–0.16	<DL-0.03	-	-	-
pH	6.9–7.5	6.9–7.5	6.5-9.5	6.5–9.0	6.5–9.5
Conductivity/μS cm^−1^	1008–1075	291–468	2500	-	-
Turbidity/NTU	2.7–5.3	0.1–0.6	1	-	-
TSS/mgL^−1^	10.5–11.5	0	-	35	*
COD_Cr_/mgL^−1^	790–955	66.9–80.1	-	125	*
TOC/mg L^−1^	188–192	35.2–98.9	without abnormal change	30	*
∑P/mg L^−1^	5.7–7.8	0.1–0.2	-	3	*
∑N/mg L^−1^	71.6–94.2	5.9–11.0	-	30	-

* The value of the parameter should be determined on the basis of the permissible pollution load at the wastewater treatment plant.

## Data Availability

The data presented in this study are available on request from the corresponding author.

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
