# Peer review of "Integration of Ultra- and Nanofiltration for Potato Processing Water (PPW) Treatment in a Circular Water Recovery System"

_membranes, 2023, doi:10.3390/membranes13010059_

Round 1

Reviewer 1 Report

In this paper, integrated ultrafiltration and nanofiltration processes for potato processing wastewater treatment were studied. Although plenty of data were presented and the work is impressive, the authors' arguments are not well supported. The paper may be accepted for publication after major revision.  

  1. More details about flat membranes (ultrafiltration and nanofiltration) should be presented in this paper, such as morphology structures, thickness, and so on.
  2. The quality (definition) of some figures should be improved. There are spelling errors in the diagram, such as "proces" in Figure 7
  3. The conclusions are complicated and should be simplified.

Author Response

Authors:

Thank you for all your comments and suggestions. Comments were very relevant and valuable for this manuscript. After applying the suggestion of the reviewer, the manuscript is definitely more valuable. The manuscript has been reviewed and corrected by a Native Speaker.

All suggested changes by the reviewer were made in Word review mode (docx) of the revised manuscript.

Reviewer 1

In this paper, integrated ultrafiltration and nanofiltration processes for potato processing wastewater treatment were studied. Although plenty of data were presented and the work is impressive, the authors' arguments are not well supported. The paper may be accepted for publication after major revision.  

  1. More details about flat membranes (ultrafiltration and nanofiltration) should be presented in this paper, such as morphology structures, thickness, and so on.

Authors: In the process, commercially available membranes were used, the specifications of which are presented in section 2.4. The structure and morphology of the membrane have not been studied. At the research planning stage, it was assumed that the description of these parameters would not bring anything to the discussion. We focused on the practical aspect of the research. The reviewer is right that it would make the publication more attractive. Unfortunately, we do not have such data. They will undoubtedly be the subject of another publication.

  1. The quality (definition) of some figures should be improved. There are spelling errors in the diagram, such as "proces" in Figure 7

Authors: All drawings and graphics have been corrected (quality and descriptions) in accordance with the requirements of the journal.

  1. The conclusions are complicated and should be simplified.

Authors: The Conclusions section has been simplified.

Reviewer 2 Report

In this paper, the authors assessed the feasibility of using a combined ultrafiltration and nanofiltration (UF/NF) membrane system for potato-processing 109 wastewater treatment. The topic, method, and results are interesting, and the manuscript provides useful information. Nevertheless, there are still some \adjustments that should be done before publishing. The detailed comments and suggestions are given below:

(1) Authors are suggested to work with a specialized commercial service to improve the fluency and readability of their paper. Your manuscript is interesting, but it’s very hard to read and contains many mistakes, thus being less likely to attract many readers and citations.

(2) The full length of each abbreviation should be noted at the first mention.

(3) Authors are suggested to elaborate on the “three membrane filtration processes” in the manuscript.

(4) It would be better if there were a mechanistic explanation for the different experimental results of these three membrane filtration processes.

Author Response

Authors:

Thank you for all your comments and suggestions. Comments were very relevant and valuable for this manuscript. After applying the suggestion of the reviewer, the manuscript is definitely more valuable. The manuscript has been reviewed and corrected by a Native Speaker.

All suggested changes by the reviewer were made in Word review mode (docx) of the revised manuscript.

Reviewer 2

In this paper, the authors assessed the feasibility of using a combined ultrafiltration and nanofiltration (UF/NF) membrane system for potato-processing 109 wastewater treatment. The topic, method, and results are interesting, and the manuscript provides useful information. Nevertheless, there are still some \adjustments that should be done before publishing. The detailed comments and suggestions are given below:

  • Authors are suggested to work with a specialized commercial service to improve the fluency and readability of their paper. Your manuscript is interesting, but it’s very hard to read and contains many mistakes, thus being less likely to attract many readers and citations.

Authors: The manuscript has been reviewed and corrected by a Native Speaker. In addition, the article has been redrafted and simplified in selected chapters (especially the Introduction section and discussion of the results). Moreover, all drawings and graphics have been corrected (quality and descriptions) in accordance with the requirements of the journal.

  • The full length of each abbreviation should be noted at the first mention.

Authors: The full lengths of each abbreviation have been added. Most of them (especially the discussed physical and chemical parameters) have been supplemented in the Materials and Methods section.

(3) Authors are suggested to elaborate on the “three membrane filtration processes” in the manuscript. It would be better if there were a mechanistic explanation for the different experimental results of these three membrane filtration processes.

Authors: In the research, we proposed an integrated UF-NF system. We indicated that it is necessary to pre-treat the raw sewage. Physical and chemical parameters of streams after individual stages were presented and discussed. We don't understand what the reviewer means by "mechanistic explanation". The article has been redrafted in selected chapters. Perhaps the issue described by the reviewer was described more clearly.

Round 2

Reviewer 1 Report

The author has responded well to the reviewer's questions, and the paper can be accepted in present form.